# *OntoDomus*: A Semantic Model for Ambient Assisted Living System Based on Smart Homes

**Hubert Kenfack Ngankam** *,†, **Hélène Pigot** † and **Sylvain Giroux** †

Laboratoire Domus, 2500 Boul. Université Sherbrooke, Sherbrooke, QC J1K 2R1, Canada;
helene.pigot@usherbrooke.ca (H.P.); sylvain.giroux@usherbrooke.ca (S.G.)
* Correspondence: hubert.kenfack.ngankam@usherbrooke.ca; Tel.: +1-819-821-8000-65-174
† These authors contributed equally to this work.

**Abstract:** Ambient assisted living (AAL) makes it possible to build assistance for older adults according to the person's context. Understanding the person's context sometimes involves transforming one's home into a smart home. Typically, this is carried out using nonintrusively distributed sensors and calm technologies. Older adults often have difficulty performing activities of daily living, such as taking medication, drinking coffee, watching television, using certain electronic devices, and dressing. This difficulty is even greater when these older adults suffer from cognitive impairments. Defining an assistance solution requires a multidisciplinary and iterative collaborative approach. It is necessary, therefore, to reason about the imperatives and solutions of this multidisciplinary collaboration (e.g., clinical), as well as the adaptation of technical constraints (e.g., technologies). A common approach to reasoning is to represent knowledge using logic-based formalisms, such as ontologies. However, there is not yet an established ontology that defines concepts such as multidisciplinary collaboration in successive stages of the assistance process. This article presents *OntoDomus*, an ontology that describes, at several levels, the semantic interactions between ambient assisted living, context awareness, smart home, and Internet of Things, based on multidisciplinarity. It revolves around two main notions: multidisciplinarity, based on specific sub-ontologies and the ambient feedback loop. *OntoDomus* combines SPARQL queries and OWL 2 models to improve the reusability of domain terminology, allowing stakeholders to represent their knowledge in different collaborative and adaptive situations. The ontological model is validated, first by its reuse in more specific works—specific to an aspect of ambient assistance. Second, it is validated by the structuring of ambient knowledge and inferences of the formalization in a case study that includes instances for a particular activity of daily living. It places the ambient feedback loop at the center of the ontology by focusing on highly expressive domain ontology formalisms with a low level of expressiveness between them.

**Keywords:** ambient assisted living; context awareness; ontologies; SPARQL; patterns; reasoning; semantic knowledge

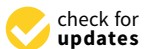



## 1. Introduction

Ambient intelligence (AmI) is a concept or multidisciplinary approach that promotes a paradigm of the technologically enhanced and smart home [1]. It offers the possibility to respond to people's everyday life activities, tasks, and needs in an easy, natural way, using information, while taking the context into account. As a domain, it is at the intersection of artificial intelligence, human–computer interaction (HCI), ubiquitous computing, context awareness, and Internet of Things (IoT) networks [2].

Activities of daily living (ADL), such as "eating", "bathing", "sleeping", or "resting", involve interactions with objects or furniture in the home. ADLs are generally used to define a person's abilities and functional status. Checking ADLs, especially for an older person, is a key factor in determining the types and levels of assistance needed and whether aging in the place is desirable. To promote independent and active aging, it is necessary to

take advantage of efforts both on the technical side and on the social side. Ambient assisted living (AAL) is an emerging multidisciplinary field aimed at providing an ecosystem of different types of wireless networks, sensors, computers, mobile devices, and software applications for personal health care monitoring and telehealth systems. AAL aims to improve the quality of life of older adults or the disabled by providing them with a safe and protected living environment [3].

To improve the quality of life of older adults, the AAL system acquires data concerning the activities of the inhabitants, their profile, their state of health, and the description of their house. Each decision is made following (1) the knowledge of the context and profile of the person; (2) data collected and understood by the sensors; (3) the interpretation and analysis of its data on the activity of the person; (4) the assistance offered to the person to accompany them during the realization of the activity. In short, AAL is built on the semantics that exist between everything that surrounds the person and accompanies them for the realization of any activity of daily life. Regarding semantics, ontologies prove to be effective and powerful tools for materializing, representing, and reasoning on the semantic aspects of knowledge. Ontologies can be seen as structured vocabularies that explain existing relations among terms, also named entities or classes [4]. They are formed by concepts and relations that can be combined to form more complex class expressions. In addition, ontologies offer the possibility of inferring new knowledge; ontological faculty would then make it possible to take into account the evolution of the clinical profile of the person.

However, the AAL challenge is to deal with a large number of elements involved in software, human, and hardware interactions to provide appropriate ambient assistance solutions. Most of the ontologies proposed in the literature focus only on technological solutions [5–9] or clinical solutions [10–14]. However, AAL is not intended just to switch devices on and off, but to bring a cognitive and clinical dimension to assisting people. AAL solutions are rich in terms of technologies but also in terms of actors and multidisciplinary approaches. This work aimed to propose a multidisciplinary approach resulting in a multidisciplinary-oriented semantic model. This model uses the underlying knowledge of technical and clinical domains to produce the inferences needed to feed the knowledge of related domains. This article mainly addresses ambient assistance comprehensively.

This work is intended to contribute to the field of domain ontologies devoted to the representation of knowledge necessary to support the realization of ADL within a smart home. The *OntoDomus* ontology uses domain ontologies to implement the AAL feedback loop. More concretely, it offers a high-level abstraction, allowing each stakeholder, either technical or clinical, to interact only with the aspect specific to their business domain. For now, it offers five important aspects of ambient assistance, as follows: (1) usage scenarios that produce user actions; (2) data captured by sensors; (3) system analysis of sensor data; (4) generation of cues to accompany the usage scenario; and (5) evaluation of user response and impact of notifications on usage scenarios. *OntoDomus* uses each of these steps to encapsulate the home description and resident profile to provide contextual ambient assistance. Thus, at each iteration of the ambient assistance loop, *OntoDomus* makes sure to structure the actions carried out by the inhabitant at each stage to restore the data collected. At the end of each step, its rules engine selects the intervention best suited to the resident's needs, and visual or auditory aids are produced to accompany the person in carrying out the activity. The actuator able to diffuse the aid is chosen according to the resident's knowledge of the environment. An iteration ends when each of the steps has been able to provide data for the next step.

The rest of this work is organized as follows: Section 2 highlights some of the research devoted to the formalization and use of ontologies in AAL. Section 3 presents the *OntoDomus* ontology, the methodology adopted, and a description of the existing semantics between the AAL entities is presented. Section 4 describes the reasoning mechanism which makes it possible to deduce the ambient knowledge. Section 5 shows a use case to describe the interaction architecture designed to allow applications to use the semantic model.

Section 6 summarizes the contributions of this article and describes the future directions of this research.

## 2. Related Work

In this section, the main concepts related to AmI, AAL system, context awareness in ubiquitous systems, ontologies that use context modeling, and IoT are presented.

### 2.1. Ambient Intelligence

Aging poses many challenges for older adults due to cognitive decline, chronic age-related diseases, and limitations in physical activity, vision, and hearing. In recent years, several studies have provided varied and diverse solutions that use assistive technologies based on a new paradigm called ambient intelligence [15].

Ambient intelligence (AmI) is a paradigm of information technologies aimed at strengthening the capacities of people through digital environments that are sensitive, adaptive, and responsive to human needs [1,16]. Several works explain the technological and infrastructural challenges to be put in place to meet this paradigm [17,18]. Mainly, works summarize the artificial intelligence methodologies used to develop an AmI system in the health field, including various learning techniques [1,2]. Some studies discuss how AmI technology could help people with various physical and mental disabilities or chronic illnesses [19,20]. This also implies proposing human–machine interactions that are characterized by ubiquitous, discreet, anticipatory, and hidden communications in intelligent environments.

### 2.2. Smart Homes

One of the goals of smart environments is to improve the quality of life in terms of comfort, safety, wellbeing, and delivering cues with efficiency. Smart environments are made up of IoT devices that work together to carry out specific operations and provide services. These environments are very diverse, including smart cities, smart homes, healthcare, and smart services [21,22]. This work focuses on smart homes.

A smart home is a standard house that has been augmented with different types of technologies. Smart home technologies include sensors, monitors, interfaces, furniture, actuators, and networked devices to enable automation as well as localized and remote control of the home environment [22–24]. Sensors and monitors detect environmental factors including temperature, light, movement, pressure, and humidity. These technologies are networked, usually using standardized IoT-based wireless communication protocols [15,23]. Rich contextual information can be obtained by analyzing and merging various types of sensor data. Most smart homes use this knowledge for automation and to provide more services to residents, as well as to assess residents' cognitive and physical health.

### 2.3. Ambient Assisted Living

Assisted living technologies based on ambient intelligence are called ambient assisted living (AAL) systems. AAL can be used to prevent, cure, reassure, secure, and improve the wellbeing and health conditions of older adults [15]. AAL systems such as medication management tools and medication reminders allow older adults to take charge of their health and reassure caregivers [25]. AAL technologies can also provide more safety for older adults, using mobile emergency response systems, fall detection systems, and video surveillance systems [15]. Other AAL technologies assist with daily activities, based on monitoring activities of daily living (ADL) and issuing reminders, as well as assistance with mobility and automation [26]. Finally, these technologies can enable older adults to better connect and communicate with their peers, as well as with family and friends. However, for optimal assistance in performing ADLs, it is necessary to understand and model the context of the delivery of the assistance.

### 2.4. Context Modeling and Ontology

Research on AAL systems has grown in importance in recent years, especially in the context modeling of smart homes. At the same time, ontology research has been exploited as a means to provide a formal, representative, and shareable conceptualization of a variety of knowledge areas related to smart homes or their components. Context modeling has received a notable amount of contributions over the past decade [9,27], ranging from simple key-value models to graph models and ontological models. As mentioned, the need for expressive modeling and support for contextual reasoning makes ontologies a primary tool for contextual modeling in many approaches, such as CONSERT, SOCAM, SOUPA, and BonSAI.

CONSERT [9] is an ontology that uses a contextual meta-modeling approach to work with domain knowledge. It defines a context representation and reasoning engine that addresses the problems of combining rules-based and ontological reasoning for domain knowledge, structured manipulation of context annotations, and detection of context integrity violations. To support rule-based event processing and reasoning, CONSERT uses a rule engine. This allows it to provide expressive context representation and ontological reasoning using semantic web technologies such as OWL and SPARQL.

To reason with multiple contexts, SOCAM, a service-oriented contextual middleware, uses ontologies to separate contextual concepts into two levels of abstraction (high-level and low-level abstraction). SOCAM's architecture proposes a formal ontology-based context model using OWL to solve problems such as the semantic representation of context, contextual reasoning, knowledge sharing, context classification, context dependency, and context quality. With these service components, SOCAM easily builds context-aware services by accessing various types of contexts with different levels of complexity.

Another study that uses a service-oriented and QoS approach is BonSAI [28]. Unlike the other ontologies, this is a domain-dependent ontology that specializes in modeling domain-specific concepts of the AmI application. To support system designers in the early stages of assistive system development [14], the authors propose an approach for the design of home automation assistance systems by presenting an ontology-based methodology aimed at guiding the development process. Each step involves the elicitation and analysis of AAL requirements and their formal representation in an ontology, where the high-level objectives are described in terms of sub-objectives and tasks, which are then linked to the corresponding measures and devices.

Based on a modular vocabulary, SOUPA [7,29] is a higher-level consensus ontology that covers aspects such as a person, time, space, events, user profiles, and user activity, ...it is increasingly used in a wider variety of management application contexts. The ontology based on a CoOL [30] model is based on the aspect–scale–context (ASC) model to link context knowledge and contextual interoperability. By using an architecture of reasoning components, CoOL produces a more abstract and global vocabulary. This vocabulary is used to facilitate transfers between arbitrary context models during the discovery and execution of services. The, it allows the model to act as a layer of interoperability and comparison. Developed by reusing widely adopted ontologies, ComfOnt [5] aims to offer residents the possibility of having personalized interior comfort in their living environment and to help them in the planning of their daily activities. ComfOnt is a semantic framework that leverages knowledge about smart house dwellers and their particular needs.

Many ambient assistive solutions have introduced their ontologies to improve knowledge interoperability in such systems. The common use of ontologies is mainly for knowledge representation, querying, and reasoning on data. In this work, although taking advantage of the representation of knowledge and reasoning, the proposed ontology infers new knowledge by ensuring to maintain a global coherence on the whole process of ambient assistance. The implementation of the ambient assistance feedback loop allows communication with sensor and actuator services, to express an ongoing state of an activity performed in a smart home. Unlike the ontologies previously presented, which directs their knowledge on a specific domain, our global modeling of knowledge between the entities of

the system makes it possible to better choose, by inference, the concepts and the relations that must intervene to offer quality assistance to ADLs.

## 3. Semantic Model of Ambient Systems

The ontology proposed in this work structures the knowledge of the field of ubiquitous computing, context awareness, the Internet of Things, and the home of the person. It enables high-level software agents to reason about facts rather than data using ontological inference engines. This section presents the construction of the ontology and defines the framework of the semantic model. It begins by defining the notion of the AAL feedback loop and the importance of receiving feedback on AAL. Then, it presents the knowledge construction methodologies used to build the ontology, and finally, it emphasizes how the proposed model is built on the existing models.

### 3.1. AAL Feedback Loop

Ambient assisted living (AAL) is focused on supporting IoT for healthcare, comfort, and control applications for home environments. Its applications consist of complex networks of heterogeneous information devices and intelligent artifacts. It often requires easy-to-install sensors, actuators, and portable devices. According to [31], the best way to detect emerging medical conditions before they become critical is to look for changes in activities of daily living (ADLs).

The aim of *OntoDomus* is to provide an interface to collect and analyze smart home data, and then to infer ADLs, performed to enable decision and action in AAL. Sensors, actuators, and an analysis system are the elements that play an important role in formulating an ADL intervention. The responsibilities of sensors are to monitor the behavior of the system and the person, while the actuators are used to act in the environment by displaying prompts that urge the person to act.

By augmenting the person's smart home with IoT, the ambient assist system can understand the context and improve the richness of the interaction, see step 1 in Figure 1. The use of IoT facilitated by calm and pervasive technologies facilitates monitoring and changing the context through continuous data collection, see step 2 in the figure. Knowing how to adapt to changing circumstances and react according to the context of use requires the use of artificial intelligence analysis techniques, as shown in step 3. Comparing the situation observed with the known situation, the system performs multilevel reasoning to find the right decision (command) to take, see step 6 in the figure. Most often in the ambient assistance device, it is the accompaniment of the elderly in the realization of the ADLs which leads to the assistance. It is therefore important in these systems to have a way to provide this assistance and to assess whether it has been received, see step 5. If the decision requires interaction with the person, then it is reflected in the environment through the actuators, see step 1 in the figure. This mechanism thus creates an AAL feedback loop, necessary for setting up the ambient assistance solution in Figure 1.

### 3.2. Methodology and Specification

The goal of this step is to abstract the model of representation and structuring of surrounding knowledge concerning the elements and entities that constitute it. we use the information and knowledge obtained through the work [32,33]. Several other ontologies [6,11,34] close to ambient intelligence were studied to find out which ones corresponded best to our needs. Inspired by the ontology development method called Methontology [35], we organized the ontology design around the four following stages: specification, conceptualization, formalization, and implementation.

- `Specification` consists of defining the objectives, the targets, and the daily uses of the ontology. To achieve this, several aspects have been addressed. For example, we can cite the types of responses to be provided by the ontology, the priority aspects to be taken into account, the main actors, and the entities of the ontology, etc.

- `Conceptualization` involves organizing knowledge step by step by defining terms, refining them, and linking them together to create semantic relationships. It also involves creating instances and implementing rules based on different concepts. More precisely, we proceeded in stages by reusing the existing ontologies, by aligning the concepts with the existing concepts. In the end, we set up the definitions of the glossary terms and define the class hierarchy.
- `Formalization` is the translation of knowledge into ontology web language (OWL).
- `Implementation` involves the use of tools to implement the formalization. The formalized concepts are expressed in RDF/OWL. In this step, we used the Protégé software. Concept assessment, verification, and validation are performed at this stage.

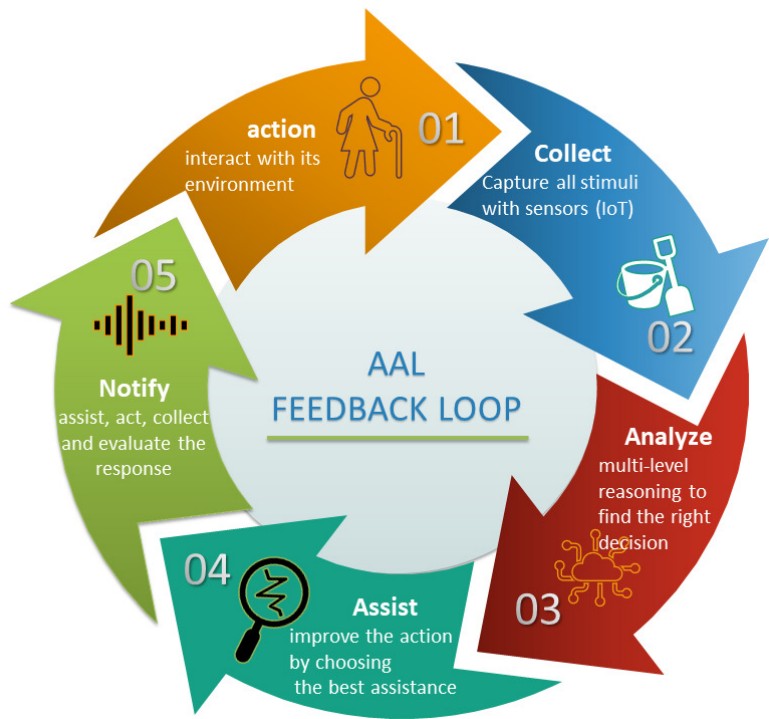

**Figure 1.** Ambient assisted living feedback loop.

To meet the needs of AAL technologies, *OntoDomus* consists of an aggregation of several ontologies. Six ontologies were designed and then assembled to form the final model, as shown in Figure 2. The "Activity" ontology models the activities of daily living from the ADL profile, as presented in this study [36]. The "Assistance" ontology deals with questions about cues tools. The "Home" ontology describes and specifies the house where people live, including relevant objects and their purpose. The "person" ontology organizes the contextual knowledge around the person including their profile. The "Task" ontology structures the concepts and the semantics of the execution and organization of tasks based on the hierarchical task analysis (HTA) model [37,38]. The "Device" ontology models the behavior and type of IoT devices (sensors/actuators/controllers/devices) required to implement a ubiquitous environment.

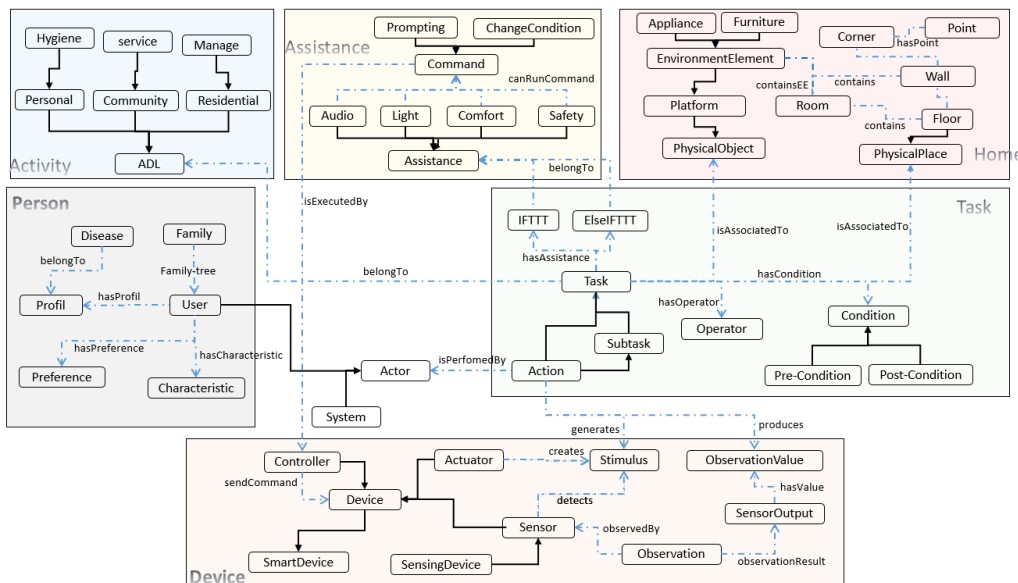

**Figure 2.** Global view of *OntoDomus* ontology.

### 3.3. OntoDomus: An Ontology for AAL

*OntoDomus*, as mentioned in Figure 2, can be decomposed into subdomain ontologies. Some of these subdomain ontologies are specializations or extensions of higher-level ontologies, such as DOLCE [39], DUL [40], and SSN [41]. The specialization of known and proven ontologies improves the rigor and extensibility of a domain's vocabulary. They make it possible to graft, in all good logic, parts of other related ontologies. The remainder of this section will describe the inner workings of each of the sub-ontologies.

#### 3.3.1. The "Home" Sub-Ontology

Concepts defined in the "Home" sub-ontology revolve around the knowledge necessary to describe an environment (habitat) and the objects that populate the habitat. It focuses on the spatial relationships between the items of the ontology. The goal is to answer the spatial questions of context awareness, including the following: Where am I? Where am I going? What can I use? What do I need? It necessitates explaining the relationship between rooms in the house. As an example, we can cite spatial relationships and spatial semantic realities, as follows:

- Are the kitchen and bathroom adjacent?
- Are the living room and kitchen separated by a corridor?
- Can you sleep in the living room?

The modeling and the demonstration of the relations between a piece of furniture or an electronic device and its place of use, its position in the habitat are concepts built in this ontology. In addition to the description of the rooms, the furniture and objects populate the rooms and their spatial relationships. It answers questions such as:

- What furniture is placed in the bedroom?
- Is the refrigerator placed near the oven?
- On which bathroom wall is the infrared sensor placed?

First of all the spatial description begins with points to build the corners and walls that make up the house. The `Point` class represents the three-dimensional coordinates of a position in space in an orthonormal coordinate system (*x*, *y*, *z*). The `Corners` class represents a corner with a precise position in the environment. As such, it is related to the `Point` class by the *hasPoints* property. This relationship reflects the fact that a corner can

only be broken down into one point. Semantically, the relationship in OWL is written as follows:

*Corners hasPoints only Points*

The `Wall` class groups the different corners of a habitat. It is therefore related to the concept of corners by the relation *hasListOfCorners*. In our model, each wall can only have exactly two corners. This definition is translated by the following relation:

*Wall hasListOfCorners exactly 2 Corners*

The Rooms class is defined as a set of two minimum walls and possibly adjacent to another room. It contains a maximum of one ceiling, one floor, and some elements of the environment (furniture, appliances), as shown in Figure 3.

```
(Contains some Appliances)
  and (Contains some Furniture)
  and (Contains some Openings)
  and (adjacentTo some Rooms)
  and (containsEnvironmentElement some EnvironmentElement)
  and (hasAssociatedScript some Scripts)
  and (Contains min 2 Wall)
  and (Contains max 1 Ceiling)
  and (Contains max 1 Floor)
```

**Figure 3.** Formal definition of a room.

### 3.3.2. The "Task" Sub-Ontology

The `Task` ontology describes and structures all the information on the organization of the actions of the person in their habitat. This ontology encapsulates the information about how a task is performed and how it could be monitored. A task is a hierarchical organization of subtasks and actions. Concerning the organization of the task, the goal is to answer questions, such as the following:

- Is the subtask "pour milk" necessary or optional in the task "make a coffee"?
- Does the "wash hands" subtask come after "flush the toilet" in the "Go to the toilet" task?
- Do you need a spoon to stir the coffee in the "make a coffee"?
- Regarding the tracking aspect, the goal is to answer questions such as the following: Which subtask is required for the success of the task?
- What are the criteria for establishing that the subtask is complete?
- Can users watch the monitor screen while performing the "take a shower" task?

Based on the script concept which describes the task flow [38], *OntoDomus* have implemented the concept of a script that encapsulates both the concept of a task, parent task, operator between task, and element of the environment. An example of script specifications is given in Figure 4.

```
(currentScript some Scripts)
  and (hasAssistance some AssistanceType)
  and (hasChild some Scripts)
  and (hasElseAssistance some AssistanceType)
  and (hasListOfAction some Action)
  and (hasOperator some Operator)
  and (hasParent some Scripts)
  and (hasPredecessor some Scripts)
  and (isAssociatedToEnvironmentElement some EnvironmentElement)
  and (isAssociatedToRoom some Rooms)
```

**Figure 4.** Formal definition of a script.

### 3.3.3. The "Device" Sub-Ontology

The `Device` sub-ontology is an extended ontology of the SSN ontology [41]. The SSN ontology was developed by the World Wide Web Consortium (W3C). It allows to describe the sensors according to their capacities, measurements carried out, observations, and deployments. As it was originally used for meteorological needs, *OntoDomus* have extended the specifications to home automation and ubiquitous systems. *OntoDomus* have added capabilities allowing them to structure a person's task in measurable and quantifiable data in their home. The Device ontology aims to answer questions such as the following:

- Does the sensor give information on people's location?
- Which controller is compatible with the infrared sensor?
- Which actuators in the kitchen may display audio signals?

The sensors of the `Device` sub-ontology monitor the stimuli produced by actions. Each stimulus converted into an observable value is associated with the output values of the corresponding sensor. Then, an abstract concept is created to allow passage of the observable value to its information-gathering process.

Referring to Figure 2, the `SensingDevice` is the class that includes all types of information gathering devices. It includes, among other things, sensors for pressures, infrared, and magnetic contacts....

The `EffectingDevice` class is the class that groups together the types of devices used to interact or give information to the person. These include lights, screens, audio speakers, and more.

The `SmartDevice` class is the class that groups so-called smart sensors. The intelligence here is defined by the communication capabilities offered by the device. It is a disjoint class of the class `UnSmartDevice`.

The `ControllerDevice` class is the class responsible for managing and representing controllers. Controllers are usually units for centralizing the collection of information in a sensor network. It also acts as a bridge between the commands sent to assist the person and the performed action.

### 3.3.4. The "Assistance" Sub-Ontology

The `Assistance` sub-ontology represents information that can be offered to a person in the AmI context. It structures the existing types of assistance and provides interfaces to send a message to a device. From the command received, it can generate high-level accompaniment corresponding to each system command. It implements the possibility of choosing, according to the profile of the person, how and under which conditions the assistance is emitted. The `Assistance` sub-ontology answers questions, such as the following:

- What is the content of the message that will be sent to report that a window has been open too long?
- According to priority assistance, is the audio mode more appropriate than the visual mode to display the message?
- For the person with memory impairment, how do you recall the time to complete the task "prepare meals"?

### 3.3.5. The "Person" Sub-Ontology

The `Person` sub-ontology is an ontology that deals with the characteristics of the person and the relationships between the people involved in AAL, including family, neighborhood, or clinicians. *OntoDomus* built the Person ontology on an existing ontology, Robert's family tree ontology [42], and extended with the clinical aspects affecting the person. It allows us to specify the concepts that are related to the clinical profile, preferences (color, music, etc.) to build ADLs specific to each person. The Person ontology aims to answer questions such as:

- Does the person have difficulty walking?

- Do their memory deficits result from an illness?
- Do they hear well?
- How does the person contact their caregiver?

### 3.3.6. The "Activity" Sub-Ontology

The `Activity` sub-ontology defines the hierarchy existing between ADL. It distinguishes self-care from more complex ADLs, called the instrumental activity of daily living. It helps to assess autonomy status. The ontology of activity aims to answer questions such as:

- Does ADL hygiene require good enough mobility?
- What ADLs require dealing with the community?
- Does shopping involve going outside?

### 3.4. Links between Sub-Ontologies

The sub-ontologies are linked together by object–property-type relationships. This form of relationship allows seamless navigation from one sub-ontology to another. For example, in the home sub-ontology, once the definition is determined, we build the existing relationships between a room and a task. The relationship *hasAvailableAction* allows linking the categories of actions that can be performed in a room.

An action is a stimulus in the environment that can produce an observable value. Completion of a task leads or begins with an action. Every time a door opens, a person moves, or an object is moved, there is a change in the environment, and these changes are translated and specified into observable values by the devices. Once *OntoDomus* has been enriched with certain instances of the the `Activity`, the `House`, and the `Person` sub-ontologies, it can generate the corresponding knowledge from the `Device` and `Assistance` sub-ontologies.

### 4. *OntoDomus* **Semantic Reasoning**

Context is the cornerstone of AmI to provide appropriate advice. Contextual reasoning aims to infer new knowledge, based on the available contextual information. As the situation of older adults changes over time, it is important to continually refresh the model of the person, including the ADLs they can perform and the assistance they need. This scalability is carried out thanks to the Jena reasoning engine [43], which injects logical inferences from atoms, axioms, and custom reasoning functions embedded in the source code. Jena provides various reasoners depending on the needs and requirements of the users. It mainly supports RDFS and OWL reasoning, but also generic reasoning. Jena's reasoner uses first-order predicate logic to perform the reasoning; its inference generally proceeds by forwarding and backward chaining. Rules of inference are usually specified using an ontology language, and often a logical description language. The Jena reasoner creates a new RDF model containing asserted and derived tuples. This extended model can be queried in the same way as a plain RDF model. *OntoDomus* develops its knowledge according to the actions carried out in the habitat and the evolution of the profile of the person. As shown in Figure 5, the Jena reasoner goes through an API to access an in-memory knowledge model. The extended model makes it possible, for example, to list the classes of the model or to determine the properties of an individual. Thus, if the reasoner determines that a specific class is unsatisfiable, the triple—`rdfs:subClassOf owl:Nothing`—will appear in the model.

Generally, Figure 5 presents the various layers that compose the semantic ambient reasoning framework and the communication between them. The design objectives of the reasoning framework are the effective representation, structuring, and dissemination of any low-level contextual information to a high level in the ambient technology dashboard. It is through this semantic structure that *OntoDomus* can generate knowledge related to other areas of ambient assistance. The inference engine iterates through the collection of

available instances and retains those that remain consistent with the instance specification of any new knowledge added.

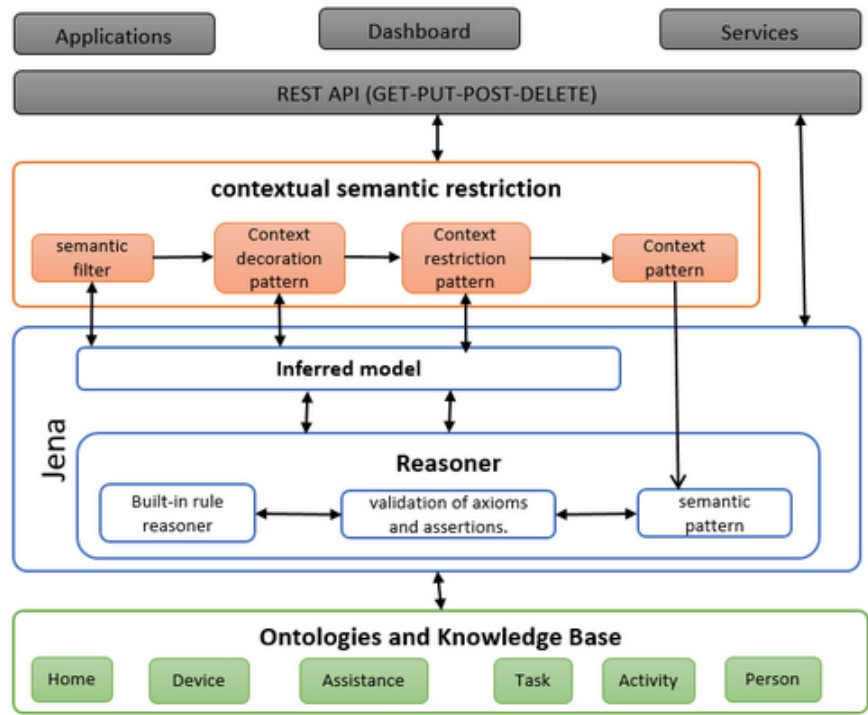

**Figure 5.** Semantic ambient reasoning framework.

Since an extended model is used, it is efficient to infer results and maintain the accuracy and completeness of the reasoning process. We carefully create the rules of Jena to prevent any result that breaks down the accuracy and completeness of the model. The lightweight and modular design of the ontology is designed for AAL applications where a large amount of data is provided by IoT devices and the habitat, thus allowing it to be processed with a small amount of knowledge to achieve good efficiency and meet resource constraints.

## 5. *OntoDomus* a Use-Case-Based Application

This ontology is mainly used to improve the semantics of the ADL execution context in a smart home. The ontology presented here was used to accompany an elderly person living alone during episodes of nocturnal wandering. Thanks to its semantic reasoning and its contextual data, it made it possible to build personalized help. It is used in an application allowing caregivers to build an intelligent environment and describe assistance scenarios.

In particular, this ontology contributes at three levels to the implementation of assistive solutions, by offering uniform semantics to the description of AAL solutions, as follows: (1) contextual filter to limit the help to be offered to the context of the person; (2) personalization of information on the home of the person with the possibility of having a high-level description of the elements of their environment; (3) semantics of the sensors with respect to what approximates the actions performed to detect an ADL.

The remainder of this section uses data from a home experiment to show how the AAL feedback loop in the *OntoDomus* ontology is modeled and used to provide assistance. For the sake of simplicity, it will address only one use case.

### 5.1. Use Case

For the evaluation of the proposed ontology, the following use case scenario, "Drinking a glass of water from the kitchen tap", was derived from a real case of an 80-year-old woman recruited in the AGE-WELL-funded project. The complete description of the participant's profile and her cognitive state is presented in the following study [44]. This

study demonstrates that the participant is often confronted with episodes of nocturnal wandering during the night and is unable to go back to bed. The objective of this ontological modeling is to conduct a qualitative feasibility study for logical reasoning in ubiquitous computing environments and to provide useful information for the implementation of contextual reasoning to help the participant drink a glass of water once she wakes up during the night. Part of the use case vocabulary is shown in Figure 6.

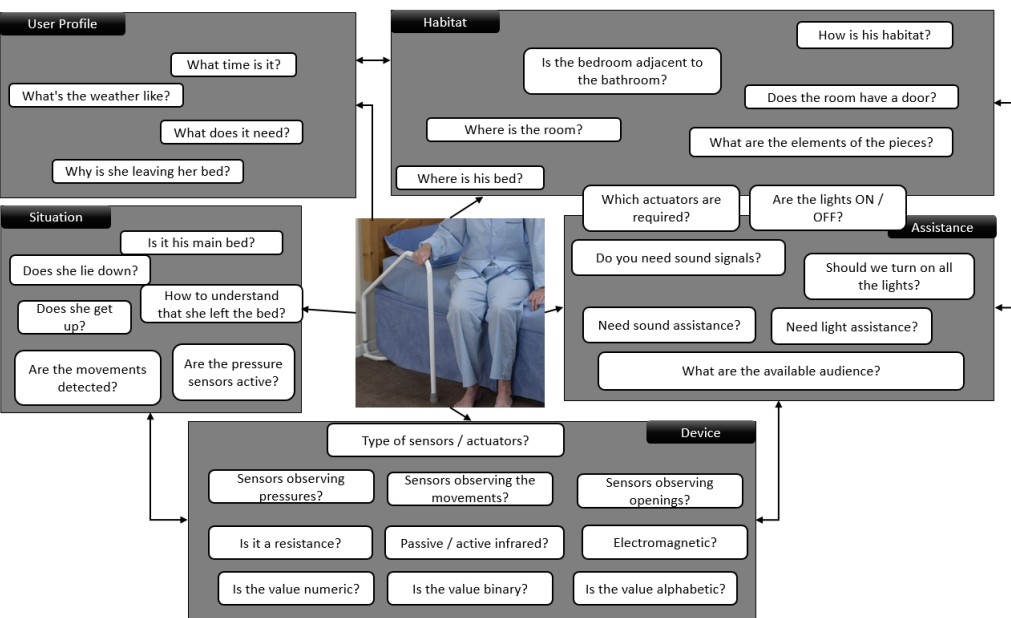

**Figure 6.** Vocabulary about getting out of bed.

Figure 6 takes into account the context to ask questions about the actions that must be performed to accomplish the scenario. At each level of execution, the semantic model provides answers to the following questions: Who? What? Where? When? How? With what? The answers are used to build context models that meet, layer by layer, the requirements of the domain ontologies used. The inquiry during the scenario completion makes it possible to group the questions into subdomains exactly as in the *OntoDomus* ontological model. Each specific question is addressed by the relevant sub-ontology, as follows:

- Why is she leaving her bed? From the user profile knowledge base;
- Is there a lamp near the bed? From the knowledge part of the habitat;
- How to detect she leaves her bed? From the knowledge base of IoT devices.

### 5.1.1. Ontological Design of the Use Case

The design requirement of the *OntoDomus* ontology is its capacity to offer a coherent and specific vision to each stakeholder, either technical or clinical, for the design of specific assistance. It was up to the semantic model to ensure the consistency and integrity of the case. More precisely, we have contextualized the coherence issues by showing a view of the situation of the scenario by subdomain of the global ontology. For each situation, we used an OWL-DL knowledge base populated with the appropriate instances to answer the questions.

Thus, the data of the profile of the person drawn from this study [44] were introduced into the ontology of the person. Sensor data was modeled using the device system subontology and stored in the knowledge base. Activities of daily living were constructed using the ADL profile model [36]. The activity knowledge base implements this aspect. The planning strategies [13] and the assistive cues tools [45] to the user are modeled in the assistance ontology. Finally, the knowledge of the habitat and the furniture of the house is structured in the sub-ontology which describes the intelligent habitat. The following study [46] shows how *OntoDomus* is used to build the smart home of Figure 7. In this

Figure, the blue blocks represent the furniture in the environment, the small green balls represent the sensors. The rounded yellow shapes represent the detection angles of the motion detectors.

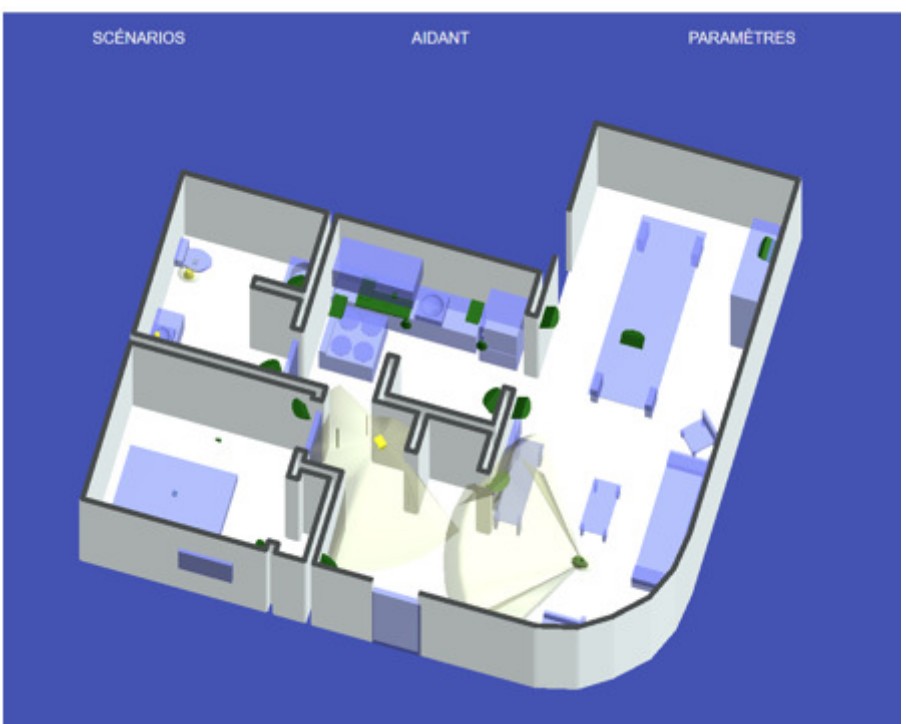

**Figure 7.** 3D representation of the smart home based on *OntoDomus*.

As *OntoDomus* is able at each iteration to reason on the semantic model produced using an inference engine (Jena's reasoner), it also ensures coherence and iterative consistency. This iterative approach allows thus to progressively validate the extended model. To represent the knowledge of the proposed use case, *OntoDomus* needs some additional concepts. Such as organization, person, and physical location in DUL. Dependency and spatial dependency, including cognitive state, are also researched in DOLCE. They have been defined as instances or specializations of the classes of DOLCE, DUL, and SSN. The results of these modelings are presented in Figure 8. It shows a partial view of the classes and instances used in *OntoDomus* to assist the drink a glass of water activity. The instances are in green in the figure, the other concepts are classes.

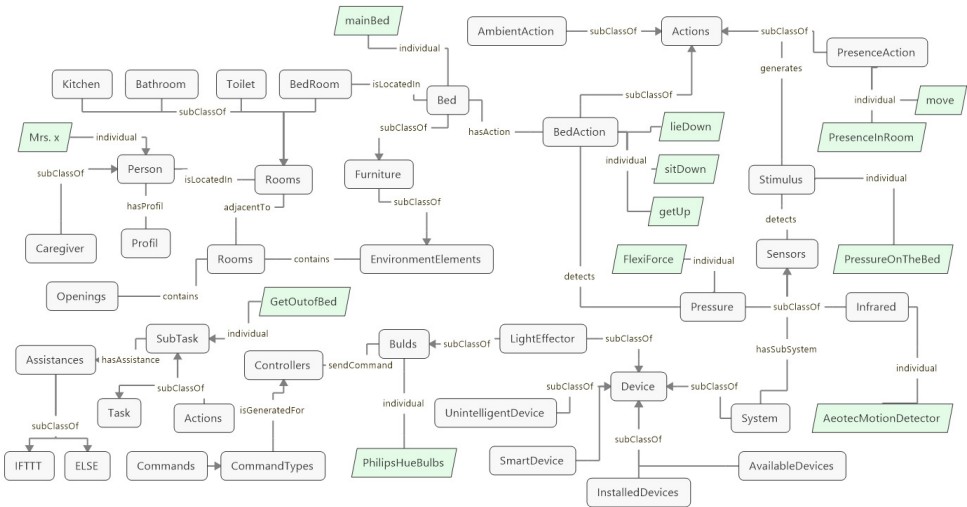

**Figure 8.** Instantiation of the *OntoDomus* for the out of bed case.

The semantic model is developed around technologies and tools: Java, Owl, Jena, Protégé, GraphDB, HTML5, and nodejs. The OWL language is used for the specification and the creation of the ontology. The Jena API is used for querying and using the ontology via the Java programming language. A REST API is built to allow the use of ontology via the HTTP and HTTPS protocol. All the design of the ontology took place in the Protégé software. No database is used, all information is structured in the ontology. The walls of the habitat, the supplies, the furniture, the appliances, the sensors, the angles of detection, the range of detection, the features of sensors, the colors, and the characteristics of each element are distributed into the *OntoDomus*.

5.1.2. Ontological Reasoning of the Use Case

This section shows how *OntoDomus* can answer general questions about assisted living. Specifically, it discusses here context reasoners built using Jena Semantic Web Toolkit, which supports rule-based inference on OWL/RDF graphs. Built-in Jena rule-based reasoners can provide semantic implications for ontologies using OWL-lite and some constructs of OWL-DL and OWL-full.

Once the semantic model of the Drinking a glass of water in the kitchen use case has been validated, the ontology instantiates in our knowledge base all the entities involved in the ambient assistance, see Figure 8. Drinking a glass of water uses two contextual reasoners (ontological reasoning based on descriptive logic and situational reasoning based on first-order logic) to perform the accompaniment. The scenario RDF graph is built and submitted to the Jena inference engine, this can be carried out by putting all the RDF data into a single model or by separating it into two components: the schema and the instance data. The methods of the Jena application, ModelFactory, in particular "ModelFactory.createInfModel", allow the reasoner to create an inference model.

After creating an inference model, all operations that access RDF statements will be able to access additional statements involved from the data bound by the reasoner. The inferred model is enriched with facts. Current facts are initial facts asserted by the user or facts derived at the current stage of resolution and facts from ontology. As an example of facts:

1. *isLocatedAt (main_bed, main_bedroom)*
2. *isDoing (Pauline, sleep)*
3. *isLocatedAt (Pauline, main_bedroom)*
4. *hasStimulus (main_bed, bed_pressure)*
5. *isAction (bed_pressure, (lie_down)∪(sit_down))*
6. *isDetected (lie_down, (pressure_mat)∪(motion_sensor) )*

The third element is the inferred knowledge of the model.

The RDF graph vocabulary consists of three disjoint sets that can be queried using a query language such as SPARQL. SPARQL improves the interoperability of applications on the Semantic Web. With Jena, it is possible to implement SPARQL endpoints that support reasoning operations. With a single SPARQL query, the inference rules are sent to the endpoint and the inferred triples are returned as the result. More complex SPARQL queries are built from the inferred model using projection (SELECT), left join (OPTIONAL), union (UNION), and constraints (FILTER). For example, the following query Figure 9 provides a list of all actions that can be performed on a bed.

```
PREFIX ssn: <http://purl.oclc.org/NET/ssnx/ssn#>
PREFIX family: <http://www.co-ode.org/roberts/family-tree.owl#>
SELECT (?action AS ?id)  ?environmentElementType ?typeEnvironment ?name ?description ?type   WHERE {
BIND (<http://www.domus.usherbrooke.ca/diy-aide/ontologies/2017/2/automation#Bed> AS ?environmentElementType).
    ?environmentElementType rdf:type ?typeEnvironment.
    ?environmentElementType domus:hasAvailableAction ?type.
    ?type rdfs:subClassOf domus:Action.
    FILTER (?type != domus:AppliancesActions && ?type != domus:FurnitureActions && ?type != domus:SanitaryActions).
    ?action rdf:type ?type.
    OPTIONAL { ?action domus:name ?name.}.
    OPTIONAL { ?action domus:hasDescription ?description. }.
}ORDER BY  ?name ?description ?type
```

**Figure 9.** SPARQL query example.

## 6. Conclusions and Further Works

This work presents *OntoDomus*, a semantic knowledge base developed to provide comprehensive contextual information to AAL applications. Validated ontologies are used to describe and represent several features of the ambient intelligence ecosystem. In the specific case presented in this article, *OntoDomus* describes how the accompaniment to the realization of the activities of the daily life of the seniors can be done. The ontology is designed to take into account the multidisciplinary management of ambient assistance. Mainly, it emphasizes the ambient feedback loop which allows successive iterations to validate the evolution of the process at each level.

Developed under Protégé, GraphDB, and Jena, the ontology exposes a REST API allowing several applications to manipulate concepts and relationships at different levels of abstractions. Thanks to the Jena reasoner, the ontology creates a new RDF model containing asserted and derived tuples. This extended model can be queried in the same way as a simple RDF model. *OntoDomus* thus develops its knowledge according to the actions carried out in the habitat and the evolution of the profile of the person.

This ontology was used for a month to iteratively build an assistance solution for the nighttime wandering episodes of a person living alone. It is used in a study where caregivers, thanks to augmented reality, build assistance scenarios and build a description of the environment for an AAL application. In the case studies of this article, the structuring of knowledge and the description of the process of inference on the data of a participant living alone makes it possible to show and validate the ontology. The results obtained in this case and the works using this ontology show a good integration that allows taking into account the multidisciplinary process to create coherent ambient assistance solutions. In particular, it is possible to observe the flexibility that the *OntoDomus* show good flexibility to describe the different contexts and acquire information about each of them. It then uses this information to adapt to the environment and improve the AAL feedback loop.

Future work foresees a set of activities aimed at improving the representation of ADLs, providing reusable models for assistive scenarios, while explicitly describing cognitive intervention plans. *OntoDomus* should offer the possibility to model, share, validate, and simulate the specific supports necessary for the realization of ADLs linked to specific cognitive disorders.

**Author Contributions:** Conceptualization, H.K.N., H.P. and S.G.; methodology, H.K.N., H.P. and S.G.; software, H.K.N.; validation, H.K.N., H.P. and S.G.; formal analysis, H.K.N.; investigation, H.K.N.; resources, H.K.N.; data curation, H.K.N.; writing—original draft preparation, H.K.N.; writing— review and editing, H.K.N. and H.P.; visualization, H.K.N., H.P. and S.G.; supervision, H.P. and S.G.; project administration, H.P. and S.G.; funding acquisition, H.P. and S.G. All authors have read and agreed to the published version of the manuscript.

**Funding:** This research was funded by AGE-WELL NCE Inc (WP 3.3, 3.3b).

**Institutional Review Board Statement:** The study was conducted according to the guidelines of the Ethics Committee of CIUSSS (Centre Intégré Universitaire de Santé et de Services Sociaux) and date of 1 June 2018.

**Informed Consent Statement:** Informed consent was obtained from all subjects involved in the study.

**Conflicts of Interest:** The authors declare no conflict of interest.

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
