# Peer review of "OntoDomus: A Semantic Model for Ambient Assisted Living System Based on Smart Homes"

_electronics, doi:10.3390/electronics11071143_

Round 1
Reviewer 1 Report
The paper presents OntoDomus, an ontology that describes at several levels the semantic interactions between ambient assisted living, context-awareness, smart home, and Internet of things based on multidisciplinarity. It places the ambient feedback loop at the center of the ontology by focusing on highly expressive domain ontology formalisms with a low level of expressiveness between them. The paper is interesting with significant scientific contributions. It can be accepted as it is.
Author Response
Thank you very much for your constructive comments.
The following parts have been improved
OntoDomus} an application based on use cases
the conclusion
Reviewer 2 Report
The introduction is good sustained by the references present in an impressive number.
I didn't see anywhere some numbers or statistical data for comparing in the relating work of ontologies study. I'm sure even in reference study are different procente presents.
The paper looks like a describe of ontology and not like a research papers with data from case study and where is present how much this ontology improve the AAL life.
The contest of software is too much describe, to say like this, then uses of it with answers of questions put in each step and in the end to present statistical values of questionaries.
For improving the paper, it needs to be present some date in comparison with new software. The conclusion needs to be sustained by real data not only by future work for validating the ontology proposed.
Author Response

(The authors gave the same response as above.)
